# A Distributed Biased Boundary Attack Method in Black-Box Attack

**Fengtao Xiang [1,*]**, **Jiahui Xu [1]**, **Wanpeng Zhang [1,*]** and **Weidong Wang [2]**

1   College of Intelligence Science and Technology, National University of Defense and Technology, Changsha 410000, China; xjh@nudt.edu.cn
2   School of Civil Engineering, Central South University, Changsha 410000, China; csuwwd@csu.edu.cn
*   Correspondence: xiangfengtao@nudt.edu.cn (F.X.); wpzhang@nudt.edu.cn (W.Z.)

**Featured Application: The paper is about black-box attack methods and the generation of adversarial samples.**

**Abstract:** The adversarial samples threaten the effectiveness of machine learning (ML) models and algorithms in many applications. In particular, black-box attack methods are quite close to actual scenarios. Research on black-box attack methods and the generation of adversarial samples is helpful to discover the defects of machine learning models. It can strengthen the robustness of machine learning algorithms models. Such methods require queries frequently, which are less efficient. This paper has made improvements in the initial generation and the search for the most effective adversarial examples. Besides, it is found that some indicators can be used to detect attacks, which is a new foundation compared with our previous studies. Firstly, the paper proposed an algorithm to generate initial adversarial samples with a smaller $L_2$ norm; secondly, a combination between particle swarm optimization (PSO) and biased boundary adversarial attack (BBA) is proposed. It is the PSO-BBA. Experiments are conducted on the ImageNet. The PSO-BBA is compared with the baseline method. Experimental comparison results certificate that: (1) A distributed framework for adversarial attack methods is proposed; (2) The proposed initial point selection method can reduces query numbers effectively; (3) Compared to the original BBA, the proposed PSO-BBA algorithm accelerates the convergence speed and improves the accuracy of attack accuracy; (4) The improved PSO-BBA algorithm has preferable performance on targeted and non-targeted attacks; (5) The mean structural similarity (MSSIM) can be used as the indicators of adversarial attack.

**Keywords:** adversarial samples; black-box attacks; machine learning models; boundary attacks

## 1. Introduction

Deep learning is occupying the core of the rapidly developing field of machine learning and artificial intelligence research. It also demonstrates good performance on many tasks, especially in the field of computer vision. However, modern deep networks are very vulnerable to adversarial samples, which presents a great threat to the effectiveness and stabilization of the community [1–5]. It is shown that image classification methods based on deep neural networks (DNN) are always fragile. They will make mistakes when only small disturbances occur, which are invisible to the human eye. This indicates that many machine learning models trained with a large number of samples are still vulnerable to small adversarial disturbances. The algorithm to find this kind of anti-interference to the input image is usually called an anti-attack. The mathematical expression is as Equation (1). The $C(x)$ is the classifier. In machine learning, the function of the classifier is to judge the category of a new input sample based on the labeled training data. Value $(x, y)$ is an input sample and its label. The attack is to produce an adversarial sample $x^{adv}$ of $x$. When $x^{adv}$ is

misclassified by ML model. The $L_p$ norm of the difference between it and the original one is below a set threshold $\varepsilon$. It means that the attack succeeded.

$$C\left(x^{adv}\right) \neq y,\ \text{s.t.} \|x^{adv} - x\|_p < \varepsilon \tag{1}$$

where $\|\bullet\|_p$ is $L_p$ norm.

The generation of adversarial samples of the image classification model is the process of adding subtle interference, which is difficult for humans to recognize in the original image. It is inputted into the deep neural network. The deep neural network methods make wrong classification results. In Figure 1, slight interferences are added to the original image. The deep neural network methods give wrong judgment. The monkey is identified as a hand sanitizer.

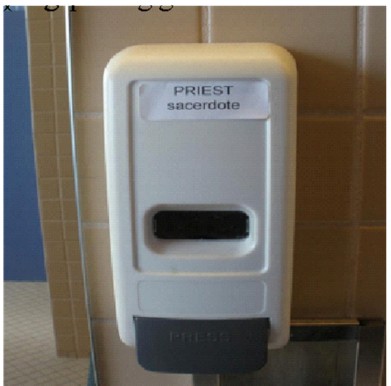
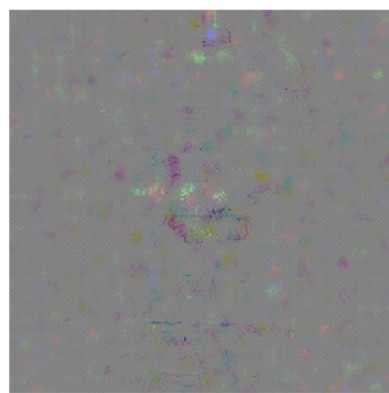

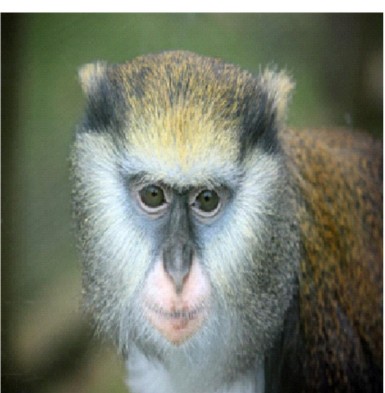
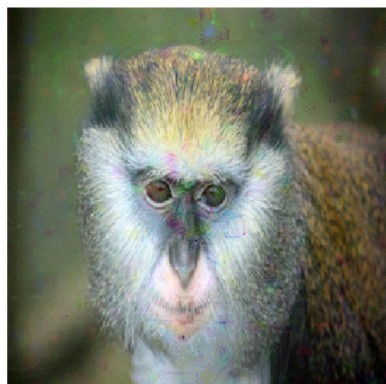

**Figure 1.** An example of an image adversarial examples attack.

Adversarial attack methods generally fall into two categories on the basis of attack effect and method. In the light of attack effects, these come in two classes: non-targeted and targeted attack methods. Non-targeted attack refers to the generation of adversarial samples by adding interference. It causes the classification algorithm to mistakenly classify the original image into images of different categories. Contrarily, targeted attack causes the classification algorithm to misclassify the original image as an image of a specified category.

On the basis of methods, these come in two classes: white and black-box methods. The white-box attack methods know about attacking deep neural network methods and model parameters in advance, such as the construction of networks, the situation of layers, neurons, etc. Additionally, they know the kinds of hyperparameters about the neural network, and the weights of connected neurons, for example. The attacker can design a targeted adversarial sample on the basis of the known information with regard to the neural network. This can make the neural network method lose efficacy. Such methods comprise the fast gradient attack proposed by Ian Goodfellow in 2014 [6] (Fast Gradient

Sign Method, FGSM), Basic Iterative Method [7] (BIM), and DeepFool [8], Jacobian-based Saliency Map [9] (JSMA), Houdini [10] and Carlini & Wagner [11], and so on. Such attack algorithms mainly rely on gradient-based attack strategies.

Conversely, black-box attack methods do not know any information about the network, such as architecture or various hyperparameters. These only have the output results of the network and corresponding data labels. These ones are closer to the real application scenario than white ones. Such methods are harder to implement than white-box methods. Meanwhile, the attack effect is relatively poor. In terms of safety, the robustness of the neural network can be improved by the research of black-box methods. The network is trained with the corresponding adversarial samples. It is very helpful in improving the security of the network application.

The black-box attack methods can be generally divided into three categories: score-based attack, transfer-based attack and decision-based attack methods. Score-based attacks rely on predicted scores, such as the class probability or distribution of the model. These attacks use predicted values to estimate the gradient, mainly including the black-box variants of JSMA [12], zero-order Optimization attacks [13], predictive adversarial generative networks [14], and simple black-box attacks [15–17]. The disadvantage of this type of method is that it is difficult to obtain the score and probability distribution of the output sample in real applications, so it cannot be practically used.

A transfer-based attack does not rely on model information. It requires information about training data, which are used to train a fully observable agent model. The adversarial disturbances can be synthesized [18]. It relies on empirical observations. The adversarial examples are often transferred between models. If adversarial examples are created on a set of proxy models, the success rate of attacking models can reach 100% in some cases [19]. Dong proposed a prior stochastic gradient free method [20] to improve the performance of black-box adversarial attacks. The algorithm also uses transfer based on the priori and score-based information. The attack success rate is higher than only one method. If the data sets are inconsistent, the attack effect may decrease a lot, and the stability is not robust [21].

Decision-based attacks are direct attacks that only rely on the final decision of the model, such as Top-1 classification labels. These attack methods are aimed at the situation where the black-box output classification results can be obtained. In an actual situation, the confidence scores or model distribution are difficult to obtain. Compared with score-based attacks, decision-based attacks are more consistent with real applications. Decision-based attacks are more robust to standard defenses than other types of attacks, such as gradient masking, inherent randomness or robustness training. Compared with transfer-based attacks, decision-based attacks require much less information about the model: neither architecture nor training data. They are simpler to apply. The disadvantage of such methods is that the queries are much higher than in other scenarios. At present, decision-based attacks mainly include the boundary attack proposed by Brendal et al. in 2018 [12] (Boundary Attack, BA) and its variants [22]. Thomas Brunner is equivalent to the biased boundary attack proposed in 2019 [13] (Biased Boundary Attack, BBA). On the basis of the attack algorithm, low-frequency disturbance and surrogate model disturbance are added. The cutting plane attack proposed by Ren [23] (Cutting plane attack) improves the convergence method based on the boundary attack algorithm. Comparative analysis of the above three types of black-box attack methods is shown in Table 1.

**Table 1.** Comparative analysis of three types black-box attack methods.

| Black-Box Attack Methods | Attack Target | Attack Conditions | Algorithm Characteristics |
| --- | --- | --- | --- |
| Score-based attack | With target/no target | The class probability or distribution of the model is required. | Use predicted values to estimate the gradient. |
| Transfer-based attack | With target/no target | Does not rely on model information, but requires information on model training data. | Train a fully observable agent model from which adversarial disturbances can be synthesized. |
| Decision-based attacks | With target/no target | Only need the output labels | The number of queries required for the attack may be higher. |

Comprehensive comparison of various black-box attack methods shows that black-box adversarial methods can accomplish targeted and non-targeted scenarios. However, in score and transfer-based methods, information on training data and models are required. Strictly speaking, they are not a complete black-box method. By contrast, decision-based boundary methods are simple and flexible concepts. They simply need to know about the labels of results. These are a type of "complete black-box attack method", and keep pace with real scenarios. They have good performance in target and non-target attacks with great application potential.

Currently, boundary attack methods and improved ones are best performance decision-based methods. A boundary attack algorithm is a black-box attack algorithm that meets the requirements of the real environment. It has superior effects in scenarios where only the input image label is obtained, and it is simple, easy to implement and robust. The improved one won the runner-up in the 2018 NeurIPS competition, named Biased Boundary Attack (BBA). It showed the BBA's superiority in actual adversarial scenarios. However, BBA still has the following two challenges: (1) query numbers are high on the premise of guaranteeing the success rate; (2) The algorithm converges slowly.

This paper studies and analyzes the advantages and disadvantages of various current black-box attack methods. The decision-based methods are the research focus. The paper combines this with the swarm intelligence algorithm based on the biased boundary attack black-box adversarial method. A biased boundary attack adversarial algorithm based on particle swarm optimization is proposed. The methods mainly have the following innovations:

Firstly, the method of initial point generation is proposed for improvement. Initial point selection has great influence on the properties of BA and BBA. The original method for selection of the initial point cannot achieve the optimal effect. This paper proposes a new algorithm for the generation of the initial attack point. The experiment shows that it can reduce the number of attack queries and the convergence speed of the algorithm.

Secondly, the particle swarm optimization algorithm is used to search the optimal adversarial samples from multiple initial points. In the framework of the BA algorithm, the generation of optimal adversarial samples can be modeled as a regional search optimization problem. The swarm intelligence algorithm has advantages of fast convergence and a good optimization effect on this kind of problem. It is feasible to introduce swarm intelligence algorithms as the direction of improvement. We try to apply the particle swarm optimization algorithm to the algorithm-updating process. Multiple initial points are adopted to find the best adversarial sample points. It can not only expand the search space of the algorithm, but also improve the convergence rate. It avoids falling into a local optimal solution effectively. It also promotes the dependability and robustness of attack applications.

With the above corrective measure, on the premise of complete black-box with only input sample query labels, the proposed algorithm achieves target/non-target attack. Compared with BA, BBA and other baseline algorithms, query numbers are reduced. The success rate of adversarial attacks is improved. The convergence speed of the algorithm is better.

The rest of the paper is organized as follows. In Section 2, the related works on Boundary Attack (BA) and Biased boundary attack algorithm (BBA) are introduced. Section 3 proposes a distributed framework for adversarial attack methods. The experimental analysis is also given in Section 4. Finally, the disadvantages and future development directions are discussed concerning future work.

## 2. Related Works

### 2.1. Boundary Attack (BA)

Boundary attack algorithm is a kind of typical decision-based method. Starting from an initial adversarial image, a binary search is used to find a sample point, which is near the classification frontier. Random walk is performed along the frontier between two opposite areas. It reduces the distance from the target image. That is to say, the classification result obtained by input classifier query is always the category we want to misclassify. According to this step, we continue to iterate and gradually reduce the distance from the original image. The reason why this kind of algorithm is named "boundary attack" is that it generates adversarial samples by searching along the boundary until it converges to obtain the optimal or satisfactory solution.

The results obtained by this kind of method can meet the requirements for misclassification of the black-box model. The overall disturbance that it increases relative to the original image varies with the performance of the algorithm. Therefore, the criterion for measuring a successful adversarial sample is that the difference $||x^{\text{adv}} - x||_p$ between it and the original image should be less than the given threshold $\varepsilon$. Take the ImageNet data set as an example: the image size is $299 \times 299$. The pixel value interval is (0, 255). When $p = 2$ (that is, the $L_2$ norm) $\varepsilon$ takes 25.89. This measure is used in our research and experimental results. As shown in Figure 2a, the generation of optimal adversarial samples can be considered as a search and optimization problem along the classification boundary between adversarial space and original space. Those boundary points that are both close to the original image and on the adversarial sample space are the feasible solutions that we want to find. However, in the decision-based black-box adversarial scene, the position and gradient direction of the real classification boundary are unknown. Therefore, it can be regarded as a high-dimensional space search problem in an uncertain environment. In the context of this complex problem, the BA algorithm adopts the method of distributing P from an appropriate pixel space. An adversarial disturbance is found closer to the original image according to a given adversarial criterion. At the same time, in order to overcome the uncertainty of the classification boundary, it needs to consume several queries to obtain a successful adversarial disturbance. The basic flow of the algorithm is as follows.

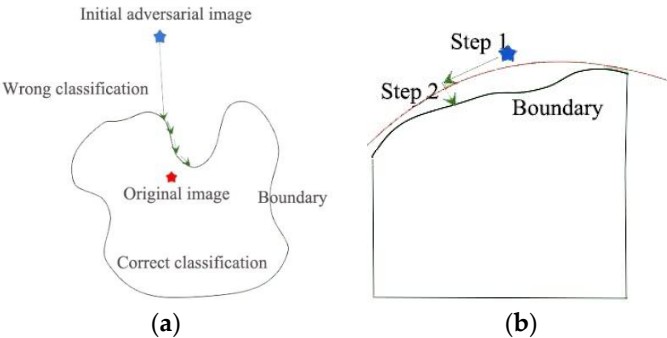

(a) (b)

**Figure 2.** Schematic diagram of boundary attack algorithm. (**a**) Initial adversarial image optimization along boundary (**b**) Move along the orthogonal direction and close to the original image.

The first step is initialization. In the non-target scene, each pixel in the initial image is sampled from the uniform distribution U (0, 255), while the non-adversarial samples are rejected. In the target scene, we start with any sample that the black-box model can recognize as the target class.

The second step is to determine the disturbance. In order to ensure that the disturbance of each sampling is improved along the direction of the classification boundary as much as possible, as shown in Figure 2b, firstly the disturbance $\eta_k$ with the sampling step size of $\delta$ is sampled on the hypersphere with the original image as the ball center, which is called the orthogonal disturbance. The disturbance of one step length of $\varepsilon$ is sampled towards the direction of the original input image. Then the next adversarial sample point is obtained.

Boundary attack is easy to use. It just needs to adjust the two parameters of step size $\delta$ and $\varepsilon$. It has a good effect on target and non-target attack. This method can find adversarial samples that are similar to the effect of gradient-based white box attacks. It does not rely on the proxy model trained on the data, which is similar to the attack model. Besides, it has stronger robustness to the common defense methods, such as gradient masking, inherent randomness and robust training. The disadvantage is high query times and the efficiency is low. The black box model needs to be queried many times to generate successful adversarial samples. A distributed particle swarm optimization (PSO) algorithm is proposed to improve the initialization and optimization process. The improved details are introduced below.

### 2.2. Biased Boundary Attack Algorithm (BBA)

However, the BA samples the perturbation direction randomly (unbiased). The number of queries is too high and the convergence speed is slow in BA. Three biased perturbation directions are used in BBA algorithm: low-frequency perturbation, regional mask and gradient of surrogate model. The purpose is to reduce the size of the disturbed sampling space by using the three kinds of prior knowledge. It is proved feasible in practice by reducing the number of queries.

### 2.2.1. Low-Frequency Disturbance

The disturbance of the original attack is sampled from the Gaussian distribution. Guo proposed a method, which has better attack properties by adding low-frequency disturbance [24]. Then Brunner improved the BA algorithm by sampling disturbance from Perlin noise distribution.

### 2.2.2. Regional Mask

The boundary attack is the interpolation from the target image to the attacked image. In some areas, these images may already be similar, while in others they are very different. By creating the region mask of the image, we can reduce the search space by taking larger disturbances in the different regions. The specific measures are below.

Firstly, an image mask $M$ is created according to the pixel difference between the adversary image and the original image.

$$M = |O_{\text{adv}} - O_{\text{origin}}| \tag{2}$$

where $O_{\text{adv}}$ is the pixel value of the adversary image. $O_{\text{origin}}$ is the pixel value of the adversary image.

Then, the mask is recalculated according to the current position at each step. It is applied to the quadrature perturbation $\eta_k^i$ of the previous sampling according to the element-wise.

$$\eta_{biased}^k = M \cdot \eta^k; \; \eta_{biased}^k = \frac{\eta_{biased}^k}{\|\eta_{biased}^k\|} \tag{3}$$

The distortion with large difference pixels is enlarged. A similar one is suppressed. Meanwhile, the disturbance vector is still unchanged.

2.2.3. Gradient of Surrogate Model

The transfer-based attack algorithm has achieved good performance in black-box attack applications. However, it will fail when the agent model is not consistent with the decision boundary of the defender closely. The transfer gradient information still works even in this case. The attack can be implemented by combining the original disturbance with limited gradient.

Firstly, the gradient of the surrogate model is calculated. Since the current position is already adversarial, a short distance is moved to the original image to ensure that the gradient is calculated from the non-adversarial region. Then the gradient is orthogonally projected to the original image direction. This projection is on the same hyperplane as the candidate points of the orthogonal step. Then, the candidate perturbations are biased to the projection gradient. After normalizing all vectors, the results are obtained below.

$$\eta_{biased}^k = (1 - w)\eta^k + w\eta_{PG}^k \qquad (4)$$

where $w$ regulates the bias strength. It is tuned-up on the basis of the property of the agent. The alienability is large, the $w$ value is increased. With above biased disturbances, sampling space has effective reduction. The query numbers are also reduced.

## 3. The Proposed Distributed Framework

### 3.1. Improvement of Initial Point Generation Algorithm

In BA methods, the generation of adversarial samples is equivalent to a search problem. One of the directions that can be improved in this problem is to define a heuristic information with a guiding function. The BBA is to introduce heuristic knowledge for the BA algorithm to speed up the search speed and efficiency. In addition, another important improvement direction is reducing the size of search space. In both BA and BBA algorithms, the approach taken begins with a single initial point. They search in a huge boundary space in a random or heuristic manner. This leads to the difficulty of too large search space, and the search gets stuck at local optimum easily. Therefore, this paper proposes the idea of combining particle swarm optimization (PSO) in the swarm intelligence algorithm to decompose the single-point search optimization into multiple points to search in their respective subspaces. Particle swarm optimization (PSO) algorithm is a random search algorithm based on group cooperation, which is developed by simulating the foraging behavior of birds. The purpose is to improve the possibility of searching for a practicable solution here. It can improve the success rate of adversarial attacks while speeding up the convergence rate. It can overcome shortcomings caused by the local optimal solution.

In this part, the initial point selection is discussed regarding the influence on the BA algorithm. Experiments have proved that the performance of the BBA algorithm has different sensitivity to the initial point selection [23,25]. Based on this, we further proposed a swarm intelligence search algorithm combined with multi-point starting.

In Figure 3, the selection of the initial point for the BBA algorithm is to find the adversarial image corresponding to the minimum $L_2$ norm. It is calculated between the adversarial image 1 and the red star (original image). It is used to perform a binary search for seeking out the point $X$. It is the start of the BA algorithm. However, $X$ is not the best start. In Figure 3, it can be seen that the boundary $Y$ of the image 3 has a larger norm. It is nearer to the red star. That the $Y$ converges to the best solution faster than $X$ is very likely. It can be believed that the initial point calculation method of BBA may not find the best starting point.

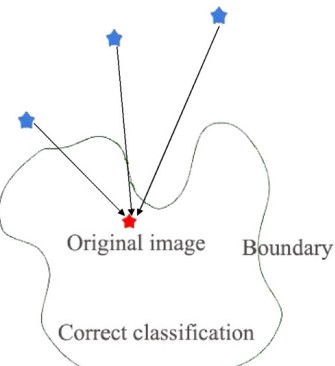

**Figure 3.** General view about the initial point of selection.

The improvement of the initial point selection algorithm is proposed. The steps are as below: At first, the boundary points are found with the origin of multiple samples, respectively. One or several (when the PSO is used) points are chosen as the start point. Theoretically, it is difficult to find the optimal method. One method is to seek out the $L_2$ norm among boundary points with the original. The closest distance is selected. The pseudo code is shown as Algorithm 1.

---

**Algorithm 1.** Improved algorithm of initial point selection

---

**Input:** Black-box classifier f, Adversarial image list $X\_adv$, Step size d, Direction $\mu$, Number of queries $t$, Threshold $\varepsilon$
**Output:** Initial adversarial image list $X\_start$

1: $d_l = d, d_r = d$;
2: for $i$ = 1 to len($X\_adv$):
3: while $f(X\_adv[i] + d_r\mu) = f(X\_adv[i])$ do
4:  $d_r = (1 + t)d_r$;
5: while $f(X\_adv[i] + d_l\mu) \neq f(X\_adv[i])$ do
6: $d_l = (1 - t)d_l$;
7: while $d_r - d_l > \varepsilon$ do
8:  $d_m = (d_r + d_l)/2$
9: if $f(X\_adv[i] + d_m\mu) = f(X\_adv[i])$ then
10:  $d_l = d_m$
11: else
12:  $d_r = d_m$
13:  $X\_start[i] = X\_adv[i] + d_r m$
14: end
15: Output initial adversarial image list.
16: return $X\_start$

---

### 3.2. BBA Algorithm with PSO Improvement

In this part, the particle swarm optimization algorithm is combined with the BBA algorithm. The advantages of multi-point starting are used to seek out the best solution. The efficiency is improved for searching the best adversarial samples. Query times are reduced with ensuring the success rate. The BBA-PSO is as below.

(1) The $X\_start$ is the start particle. The *candidate_list* is the initial velocity;
(2) The particle $i$ fitness function is below.

$$fitness_i^k = l_2\|X_i^k - X_{origin}\|$$ (5)

where $k$ is iteration numbers, $X_{origin}$ is the original image.

The update formula of velocity is:

$$V_i^{k+1} = wV_i^k + c_1r_1\left(P_i^k - X_i^k\right) + c_2r_2\left(P_g^k - X_i^k\right) \tag{6}$$

where $V_i^{k+1}$ is velocity of next iteration and $\omega$ is a parameter of inertial velocity. $P_i^k$ is the current optimal position. $X_i^k$ is a position. $P_g^k$ is the current best position. $c_1$ and $c_2$ are acceleration coefficients, which are used to control the influence on the flight direction of particles. $r_1$ and $r_2$ are independent random numbers between [0, 1]. The characteristics of PSO are reflected which can keep the previous direction of velocity. Simultaneously, it also moves towards the best orientation. For the BBA problem, PSO can be used to update the disturbance, which retains the current optimal disturbance direction and moves towards the global optimal direction at the same time. The updated formula of position is as below:

$$X_i^{k+1} = X_i^k + V_i^{k+1} \tag{7}$$

In adversarial environments, the perturbation space is searched and optimized by the continuous update of the particle swarm. When the updating position is outside the scope of sample space, the perturbation created by Equation (3) is the velocity updating to guarantee that the particle position stays in it. The procedure is shown in Figure 4 of PSO to seek out the best sample. The blue star is searched as the original particle. The green arrowhead is the disturbance of PSO updating by the iteration in Equation (6). The process does not waste any number of queries. Instead, it updates autonomously on the basis of the current orientation of each particle, and the best one. The query is needed for each update to judge only once that a particle has exceeded the boundary or not. If it is judged to exceed the space, a generated velocity is conducted as the correction by Equation (3). As the red arrowhead shows, the best sample is finally found through search, that is, the position of the yellow star. The adversarial sample search problem essentially has multiple solutions. Sometimes, it only needs to seek out one solution quickly. It can then be considered successful. The PSO algorithm based on multiple initial points can seek out the optimization among subspaces parallelly, which accelerates the speed of solution. One of the cost tradeoffs involved is that, per particle, it has to use up certain query numbers. The determination of particle numbers and its priority requires lots of experimental research. In the experiment of this article, the initial particle numbers are taken as five. Firstly, the priority is determined synchronously. Then it is updated asynchronously. The PSO_BBA is shown as Algorithm 2.

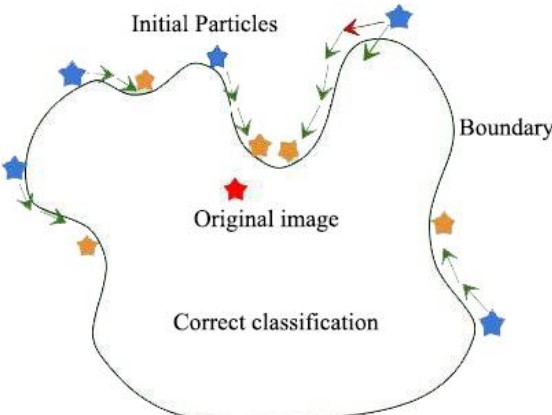

**Figure 4.** The process of particle swarm searching for optimal adversarial examples.

## 4. Experiments and Analysis

This paper evaluates the effectiveness of the PSO_BBA on dataset ImageNet comparatively [26]. It consists of 1000 classes with $299 \times 299$ size. Compared with the baseline algorithm, this paper adopts the same experiments and hyperparameter settings as those

published in previous studies [20]. The selected attack object model is a pre-trained InceptionV3 network. In order to ensure the effectiveness of the attack, the algorithms automatically skipped over the images that could not be correctly recognized by the model itself.

In the experiments, 100 images are selected randomly from the verification dataset to attack. The average $L_2$ distance is calculated for evaluating the effectiveness. The comparative methods attack on the same images to guarantee the effectiveness at the same time because the non targeted attack is relatively easy. This paper mainly focuses on the targeted attack task. The experimental results are shown in Table 2. It can be seen that our method needs less queries than BBA and BA for successful attacks.

---

**Algorithm 2.** BBA algorithm based on PSO (PSO_BBA)

---

**Input:** Initial adversarial image list $X\_start$, List of sampling disturbances $candidate\_list$, The original image $X_{origin}$, Number of particles $num\_particles$
**Output:** Optimal adversarial image $X\_adv\_best$

1:     for $i = 1$ to len($X\_start$):
2:     Initial position $X_i$, velocity $V_i$:
3:     $X_i = X\_start[i]; V_i = candidate\_list[i]$;
4:     Calculate the fitness function value and set the current local optimal position:
5:     $fitness_i^k = l_2 \| X_i^k - X_{origin} \|$;
6:     $pBest_i = X_i$;
7:     Calculate the global optimal location:
8:     $gBest = min\{pBest_i\}$;
9:     while $query <= max\_queries = 15{,}000$:
10:    for $i = 1$ to $n$:
11:      $V_i^{k+1} = wV_i^k + c_1 r_1 \left( P_i^k - X_i^k \right) + c_2 r_2 \left( P_g^k - X_i^k \right)$
12:      $X_i^{k+1} = X_i^k + V_i^{k+1}$
13:    if $X_i^{k+1} == is\_adversarial$
14:       $fitness_i^k = l_2 \| x_i^k - x_{origin} \|$;
15:    if $fitness_i^k \leq pBest_i$
16:       $pBest_i = fitness_i^k$;
17:    else $X_i^{i+1} = X_i + h_{biased}^k$
18:       $V_i^{i+1} = h_{biased}^k$;
19:    $k = k + 1$;
20:    $gBest = min\{pBest_i\}$;
21:    end
22:    Output optimal adversarial image:
23:    return $X\_adv\_best = gBest$

---

**Table 2.** Comparison of the average query numbers.

| Attack Methods | Average Number of Queries on Successful Attack (Per Image) |
| :---: | :---: |
| BA | >10,000 |
| BBA | 5432 |
| **Proposed PSO_BBA** | **4596** |

For details, the proposed method with BBA in the iteration step is compared. Figure 5 gives the variation of the average distortion relative to the original image with the number of queries. The ordinate decreases faster, the algorithm converges quicker. As shown in the figure, the improved algorithm needs to spend more queries to select the initial point for each particle in the early stage. As long as the initialization process is completed, the algorithm convergence is faster. The result is pretty well comparative to benchmarks. A comparative case of the specific display of the adversarial attack process is shown in

Figure 6. The proposed PSO-BBA and BBA algorithms are, respectively, applied on the same original image to better compare and show the improved effect of the algorithm in the paper. The experimental results show that the proposed PSO-BBA algorithm can generate adversarial samples more effectively.

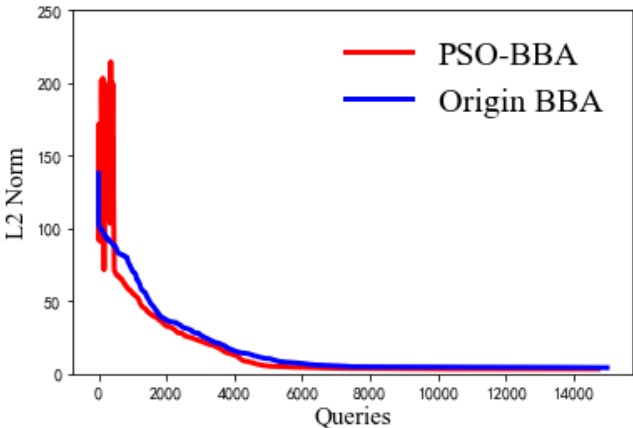

**Figure 5.** Comparison of the average distortion (L2 distance) in each iteration between the proposed PSO-BBA and the original BBA algorithm.

In addition to the $L_2$ distance, this paper also uses the mean structural similarity (MSSIM) [27] to measure the similarity between the attacked image and the original one. MSSIM is a function of evaluating image fidelity based on human visual characteristics.

$$MSSIM = \frac{1}{L}\sum_{i=1}^{L}[l(f,\hat{f})]^{a}g[c(f,\hat{f})]^{b}g[s(f,\hat{f})]^{g} \tag{8}$$

where $L$ is the number of image blocks, $l(\bullet)$, $c(\bullet)$ and $s(\bullet)$ represent the luminance, contrast and structural contrast functions, respectively. $\alpha$, $\beta$ and $\gamma$ are the weight parameters to regulate the relative importance of the three components. MSSIM mainly considers the structure information of the image itself adequately, with a value range of [0, 1]. The closer the value of MSSIM is to 1, the higher the fidelity between images used to compare. In the experiment, the values of the relevant parameters of the evaluation index are recommended by the original paper [24].

A specific black-box attack example comparison experiment is shown in Figure 7. It can be seen that the improved PSO-BBA algorithm can effectively reduce the number of queries while ensuring the success rate of the attack. Although the similarity between the initial interference image generated by each particle and the original is poor. By expanding search space, the improved PSO-BBA algorithm generates a higher degree of similarity to the original image based on all particles. It also improves the convergence speed of the algorithm in comparison to the original method. The local optimal situation is avoided effectively. The performance of the attack algorithm is also improved. Figure 8 shows an example of a failed BBA attack. The improved PSO-BBA algorithm succeeds and the number of attack queries is significantly reduced. However, the comparison results of MSSIM show that the two methods have changed the structural information of the original image. So, some evaluation indexes can be used as the indicators of detecting attacks.

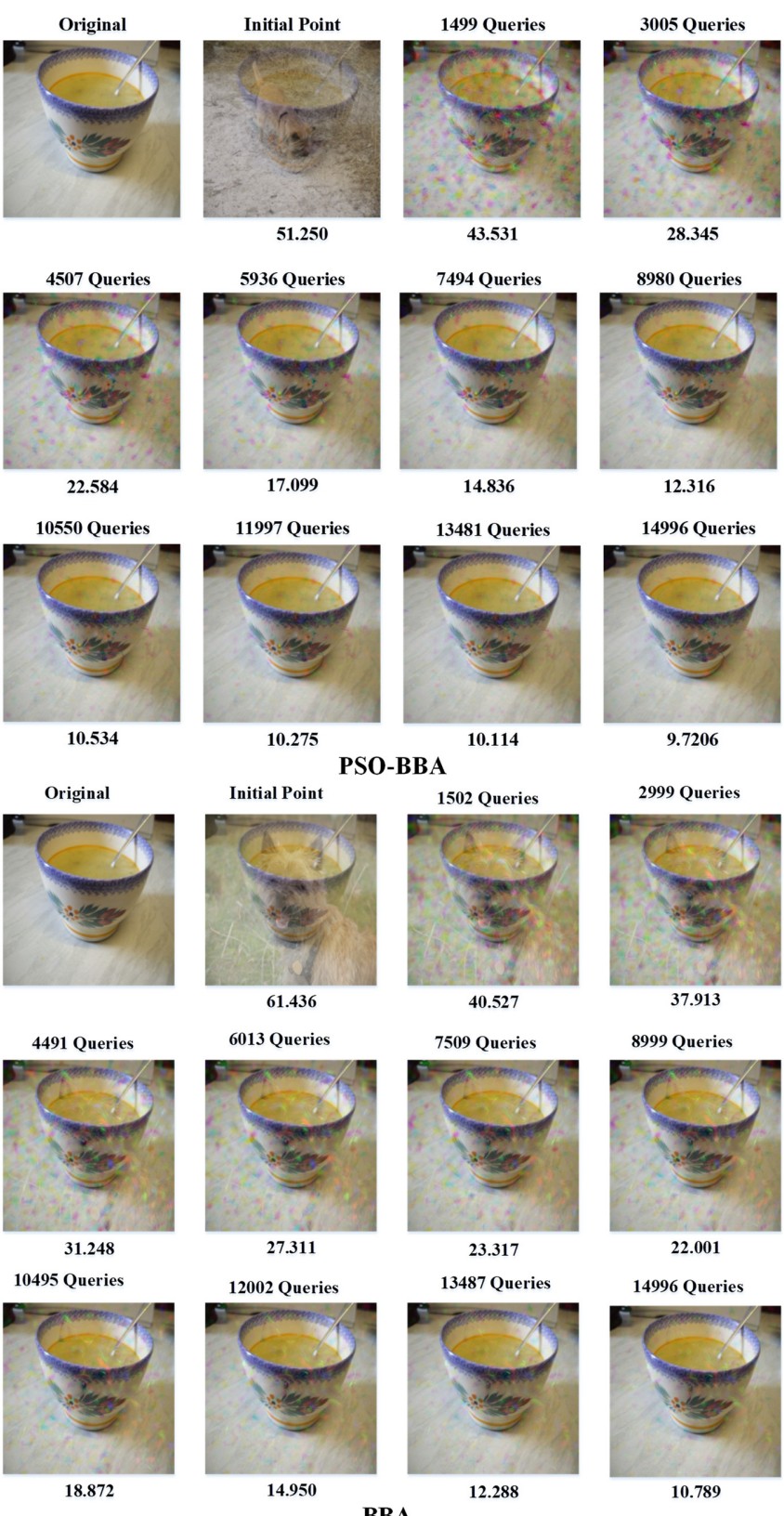

**Figure 6.** Comparison results of PSO-BBA (**top**) and BBA (**bottom**) generated adversarial examples (Query times and the distortion ($L_2$ distance)).

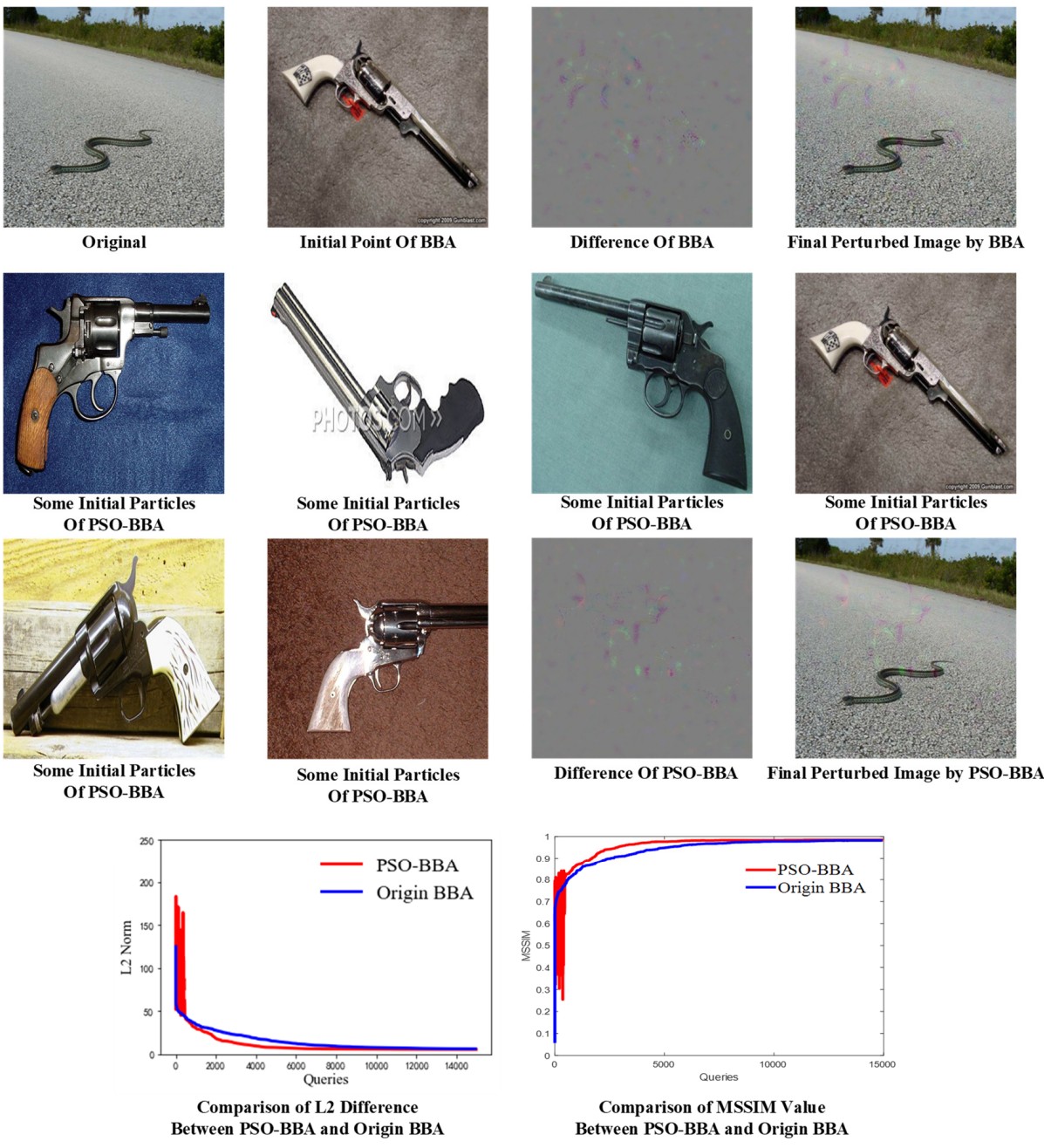

**Figure 7.** Comparison Experiment 1 of specific adversarial process and results between PSO-BBA and BBA.

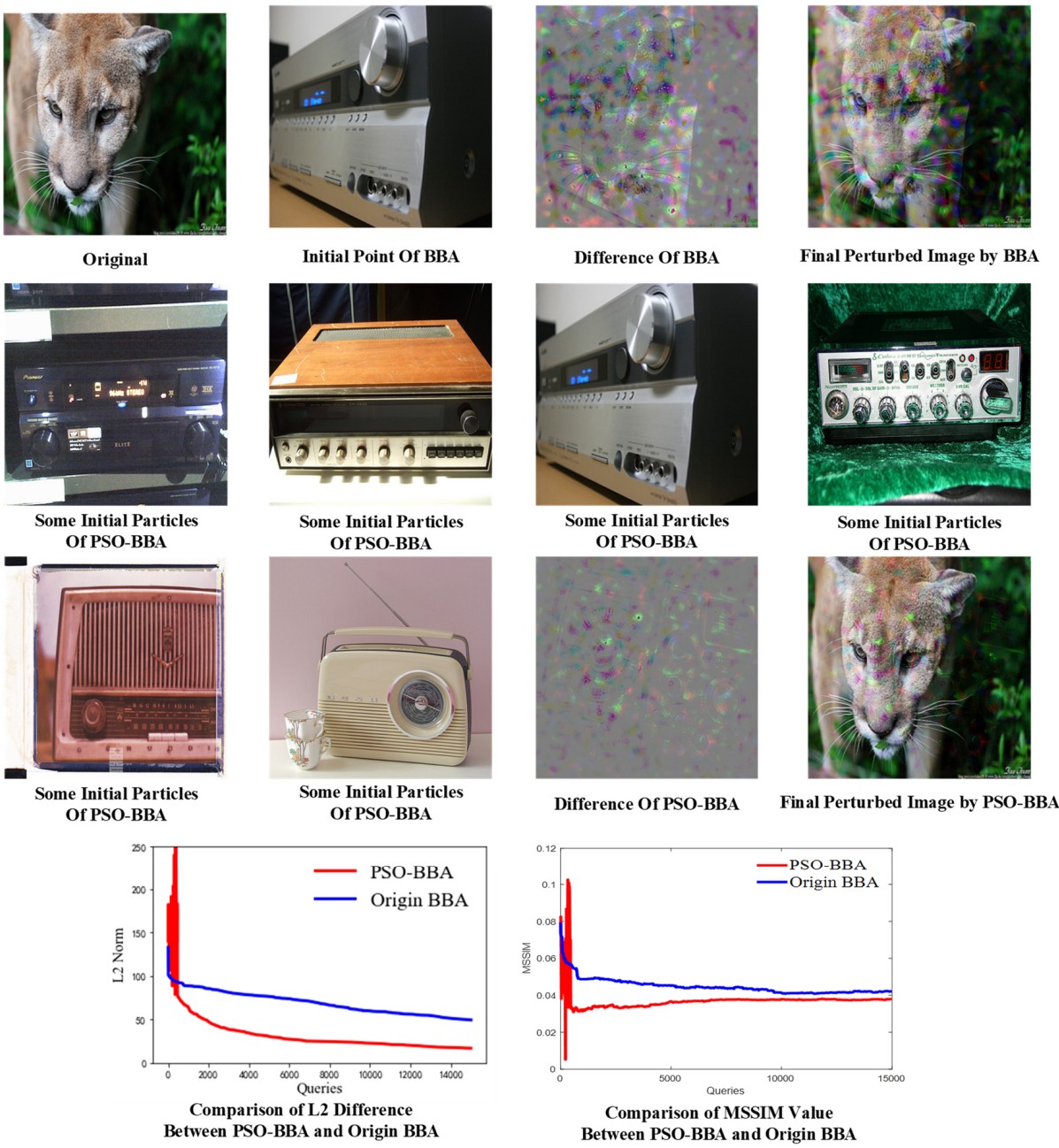

**Figure 8.** Comparison Experiment 2 of specific adversarial process and results between PSO-BBA and BBA.

## 5. Conclusions and Future Work

Aiming at the black-box adversarial attack application, hackers can obtain given input labels only. The paper researched existing problems in the black-box attack method. The PSO-BBA is put forward, which avoids the current BBA method falling into the situation of a local optimal solution. The proposed PSO-BBA method improves the convergence speed of the BBA and ensures the performance of the black-box attack. Through the experimental comparison with the baseline method, it is proved that the proposed PSO-BBA effectively reduces query numbers. It produced adversarial samples at a lower self-cost and faster efficiency successfully. Our next research will focus on conducting detailed theoretical and experimental analyses on the selection of initial particles in the improved algorithm [28].

**Author Contributions:** Conceptualization, W.Z.; Methodology, F.X.; Project administration, W.W.; Software, J.X. All authors have read and agreed to the published version of the manuscript.

**Funding:** This work was supported by the National Natural Science Foundation of China (No. U1734208, 61806212) and Natural Science Foundation of Hunan Province (No. 2021JJ40693, 2019JJ50724).

**Institutional Review Board Statement:** Not applicable.

**Informed Consent Statement:** Not applicable.

**Acknowledgments:** The authors are very grateful to the reviewers for their valuable and constructive feedbacks on this paper.

**Conflicts of Interest:** The authors declare no conflict of interest.

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
