# Peer review of "A Distributed Biased Boundary Attack Method in Black-Box Attack"

_applsci, doi:10.3390/app112110479_

Round 1

Reviewer 1 Report

The paper proposes and then compares a method to generate a black box adversarial method of attack, by combining two existing adversarial attacks. The validation method showed the effectiveness of this method in: reduction of query numbers, a higher convergence speed and higher attack accuracy, existing method of attacked vs original image comparison can detect this attack. The study presented is interesting, well written and structured. 

Author Response

Reviewer 1

Comment1:The paper proposes and then compares a method to generate a black box adversarial method of attack, by combining two existing adversarial attacks. The validation method showed the effectiveness of this method in: reduction of query numbers, a higher convergence speed and higher attack accuracy, existing method of attacked vs original image comparison can detect this attack. The study presented is interesting, well written and structured.

Explanation: Thanks for the reviewer’s comment. We tried our best to improve the manuscript and made some changes in the manuscript. These changes will not influence the content and framework of the paper. And we marked them in red in revised paper. We appreciate for Editors/Reviewer’s hard work earnestly, and hope that the correction will meet with approval. Once again, thank you very much for your comments and suggestions.

Reviewer 2 Report

This paper deals with an exciting topic. The article has been read carefully, and some crucial issues have been highlighted in order to be considered by the author(s).

• All the acronyms should be defined and explained first before using them such that they become evident for the readers.

• Most of the typos and incorrect grammars have been corrected, but it is still necessary to subject the paper to proofreading.

• The paper needs to be restructured in order to be precise.The Introduction and related work parts give valuable information for the readers as well as researchers. In addition recent papers should be added in the part of related work.

• As it is real time application oriented, authors should care over the outcome of the proposed framework by meeting the future requirements too.

• Representation of figures needs to be improved.

• Grammatical errors should be validated.

• It would be good if similar domains, such as adversarial examples, would be reflected in future research or related work.

[1] KKwon, Hyun, et al. "Classification score approach for detecting adversarial example in deep neural network." Multimedia Tools and Applications 80.7 (2021): 10339-10360.

[2] Kwon, Hyun, Hyunsoo Yoon, and Daeseon Choi. "Restricted evasion attack: Generation of restricted-area adversarial example." IEEE Access 7 (2019): 60908-60919.

[3] Kwon, Hyun, Hyunsoo Yoon, and Ki-Woong Park. "Acoustic-decoy: Detection of adversarial examples through audio modification on speech recognition system." Neurocomputing417 (2020): 357-370.

Author Response

Reviewer 2

This paper deals with an exciting topic. The article has been read carefully, and some crucial issues have been highlighted in order to be considered by the author(s).

Comment1: All the acronyms should be defined and explained first before using them such that they become evident for the readers.

Explanation: Thanks for the reviewer’s advice. We are sorry that some acronyms were not defined and explained in the first appeared. We have revised such mistakes by read-through the paper many times carefully in this revised manuscript. And we marked them in red in revised manuscript.

Comment2: Most of the typos and incorrect grammars have been corrected, but it is still necessary to subject the paper to proofreading.

Explanation: Thanks for the reviewer’s advice. The comments are all valuable and very helpful for revising and improving our paper. We have revised some incorrect grammar mistakes by read-through the paper many times carefully in the revised edition.

Comment3:The paper needs to be restructured in order to be precise. The introduction and related work parts give valuable information for the readers as well as researchers. In addition, recent papers should be added in the part of related work.

Explanation: Thanks for the reviewer’s patience. Based on our understanding, the manuscript has been relatively precised. Some irrelevant contents were not included in the article. We hope the reviewers can understand.

Comment4: As it is real time application oriented, authors should care over the outcome of the proposed framework by meeting the future requirements too.

Explanation: Thanks for the reviewer’s patience. The future requirements mainly focus on intelligent information security. Our next research will focus on conducting detailed theoretical and experimental analysis on the selection of initial particles in the improved algorithm. It is mentioned in the revised edition.

Comment5: Representation of figures needs to be improved. Grammatical errors should be validated.

Explanation: Thanks for the reviewer’s advice. The comments are all valuable and very helpful for revising and improving our paper. Representation of figures has been improved in the revised edition. Some grammatical errors have been validated by read-through the paper many times carefully.

Comment6:It would be good if similar domains, such as adversarial examples, would be reflected in future research or related work.

[1] Kwon, Hyun, et al. "Classification score approach for detecting adversarial example in deep neural network." Multimedia Tools and Applications 80.7 (2021): 10339-10360.

[2] Kwon, Hyun, Hyunsoo Yoon, and Daeseon Choi. "Restricted evasion attack: Generation of restricted-area adversarial example." IEEE Access 7 (2019): 60908-60919.

[3] Kwon, Hyun, Hyunsoo Yoon, and Ki-Woong Park. "Acoustic-decoy: Detection of adversarial examples through audio modification on speech recognition system." Neurocomputing417 (2020): 357-370.

Explanation: Thanks for the reviewer’s advice. We are sorry that it's not comprehensive enough. Adversarial examples are reflected in abstract and keywords. It is a very important aspect of our future research. In the future, we will focus on the research content of the given references. The paper 1 is added to our references.

Reviewer 3 Report

The present paper deals with a very actual and interesting domain, namely black-box attack methods and the generation of adversarial samples using ML techniques.

The state of the art of the paper is based on 27 references, most of them relatively new. taking into consideration the dynamics of the domain, it might be an asset to include studies from 2021 as well.

By reading the material, I observed are several ambiguities, as follows

- staring from the beginning, (row 41) please explain the role and physical interpretation of the classifier C(x), and of its argument.

- the phrase " The C(x) is classifier. (x, y) is input label " does not make sense. The dot is misplaced.

- what is the significance of xadv? On my opinion, starting the paper abruptly with eq(1) may confuse the reader, who should de at least become familiar with the problem and the notations used in the paper.

- again, the phrase  " When xadv is misclassified by ML model. " does not make sense - it is unfinished.

- in eq (1), who is y, what is the significance of the operator ||  ||p ?

- in Figure 1 - please explain what represents the subfigures a,b,c,d? Is there a connection between them? Is there a connection between those figurea and eq(1)?

- The phrase " The attack success rate is higher than that of a single  method. " - again, does not make much sense? what is a single method? it has not defined in text before.

- In Table 1. Comparative analysis of three types black box attack methods - the information are taken from different reference sources. Please indicate the references at each line / column of the table.

- please explain the phrase " It has superior effects in scenarios where only the input image label is obtained, " (lines 125-126) What do you understand by image label? It has not been previously defined.

- please explain what swarm / particle swarm algorithms means. Eventually give references.

- please introduce references for known algorithms - like boundary attack algorithm, random walk, Biased boundary attack algorithm.

- If Figure 2. Schematic diagram of boundary attack algorithm has been taken or inspired from existing literature - please indicate de source .

- also, if equations (2,3,4) are taken or inspired from existing literature - please indicate de source. Moreover, eq(3) is not properly aligned.

- For Table 2.  Improved algorithm of initial point selection, please explain in the text the significance of each variable used.

- in eq(6) - who are omega, c1, r1, c2, r2?

- For Table 3. BBA algorithm based on PSO (PSO_BBA) - Same fo please explain in the text the significance of each variable used.

- the images shown in Figure 5. Some examples of ImageNet dataset - are bearly visible. If they are not important (just some examples) there is no need to represent them. If they are important then enlarge the figure and emphasize those images that are important.

- same for Figure 7. Comparison of PSO-BBA (top) and BBA (bottom) generated adversarial examples (Query times and the distortion (L2 distance)). There are the results of your algorithm - explain them in text and emphasize the difference between the 24 images represented. Same for figure 6, explain the results more thoroughly in text.

- Same comment for Figures 8 and 9. It would be also beneficial to compare the results obtained by your algorithm with similar ones from existing literature, validating thus the results.

-in the conclusion section please emphasize once again the advantages and disadvantages of your approach, also in comparison with similar work presented in the existing literature.

Al last but not at least please refine your English (grammar and style). The phrases are sometimes very short - sounding clipped. The grammar should be revised. There are even some funny expressions like " In the light of […], it come in two flavors:"(rows 54 and 60). Flavor refers to taste, not classification.

A native English proof-reader might be of real help if you want to improve the intelligibility and quality of your work.

Author Response

Reviewer3

The present paper deals with a very actual and interesting domain, namely black-box attack methods and the generation of adversarial samples using ML techniques.

Comment1: The state of the art of the paper is based on 27 references, most of them relatively new. taking into consideration the dynamics of the domain, it might be an asset to include studies from 2021 as well.

Explanation: Thanks for the reviewer’s advice. The comments are all valuable and very helpful for revising and improving our paper. Some studies from 2021 is taken into consideration.

 By reading the material, I observed are several ambiguities, as follows

Comment2: - staring from the beginning, (row 41) please explain the role and physical interpretation of the classifier C(x), and of its argument. - the phrase " The C(x) is classifier. (x, y) is input label " does not make sense. The dot is misplaced.  - what is the significance of xadv? On my opinion, starting the paper abruptly with eq(1) may confuse the reader, who should de at least become familiar with the problem and the notations used in the paper.  - again, the phrase  " When xadv is misclassified by ML model. " does not make sense - it is unfinished.

 - in eq (1), who is y, what is the significance of the operator ||  ||p ?

Explanation: We are very sorry that we didn’t expatiate on the problems proposed above clearly and accurately in our original manuscript. We thank for the reviewer’s patience. They are explained in the revised manuscript and marked red. We try our best to expatiate the comment and avoid above ambiguities.

Comment3: - in Figure 1 - please explain what represents the subfigures a,b,c,d? Is there a connection between them? Is there a connection between those figurea and eq(1)?

Explanation: We are very sorry that we didn’t expatiate on the problems proposed above clearly and accurately in our original manuscript. We thank for the reviewer’s patience. a,b,c,d in Figure 1 are an example of an image adversarial examples attack. b+c=d. The monkey is identified as a hand sanitizer, which means C(d)=a. It is very relevant between Figure 1 and eq(1).

Comment4: - The phrase " The attack success rate is higher than that of a single  method. " - again, does not make much sense? what is a single method? it has not defined in text before.

Explanation: We are very sorry that we didn’t expatiate on the problems proposed above clearly and accurately in our original manuscript. We thank for the reviewer’s patience. The algorithm also uses transfer based on the priori and score-based information. The attack success rate is higher than only one method. Single method means that one method among score-based attack, transfer-based attack and decision-based attack methods rather without combination. It is expatiated in line82-83.

Comment5: - In Table 1. Comparative analysis of three types black box attack methods - the information is taken from different reference sources. Please indicate the references at each line / column of the table.

Explanation: Thanks for the reviewer’s advice. At above three paragraphs are cited references separately. Reads can find references from it. We think that it is unnecessary to quote again in the table.

Comment6:  - please explain the phrase " It has superior effects in scenarios where only the input image label is obtained, " (lines 125-126) What do you understand by image label? It has not been previously defined.

Explanation: Thanks for the reviewer’s advice. We are very sorry that we didn’t expatiate on the problems proposed above clearly and accurately in our original manuscript.  It means that the boundary attack method and improved algorithms has well attack performance only with input image labels. Image label is given from dataset. The output is also an image label. If the input and output labels is different, the attack is succeeded.

Comment7:  - please explain what swarm / particle swarm algorithms means. Eventually give references.  - please introduce references for known algorithms - like boundary attack algorithm, random walk, Biased boundary attack algorithm.

Explanation: Thanks for the reviewer’s advice. Particle swarm optimization algorithm is a random search algorithm based on group cooperation, which is developed by simulating the foraging behavior of birds. The comments are all valuable and very helpful for revising and improving our paper. The problems proposed above has been revised in resubmitted manuscript and marked red.

Comment8:  - If Figure 2. Schematic diagram of boundary attack algorithm has been taken or inspired from existing literature - please indicate de source.

 - also, if equations (2,3,4) are taken or inspired from existing literature - please indicate de source. Moreover, eq(3) is not properly aligned.

Explanation: Thanks for the reviewer’s advice. We are very sorry that we didn’t expatiate on the problems proposed above clearly and accurately in our original manuscript. Above comments are not taken or inspired by others. They are sketch map and unnecessary to indicate de source.

Comment9: - For Table 2.  Improved algorithm of initial point selection, please explain in the text the significance of each variable used.  - in eq(6) - who are omega, c1, r1, c2, r2? - For Table 3. BBA algorithm based on PSO (PSO_BBA) - Same fo please explain in the text the significance of each variable used.

Explanation: Thanks for the reviewer’s advice. The comments are all valuable and very helpful for revising and improving our paper. Each variable used in Table 2 and 3 is explained in the header of the table. c1 and c2 are acceleration coefficients, which are used to control the influence on the flight direction of particles. r1 and r2 are independent random numbers between [0,1].

 Comment10:- the images shown in Figure 5. Some examples of ImageNet dataset - are bearly visible. If they are not important (just some examples) there is no need to represent them. If they are important then enlarge the figure and emphasize those images that are important.

Explanation: Thanks for the reviewer’s advice. The comments are all valuable and very helpful for revising and improving our paper. Figure 5 is deleted in resubmitted manuscript.

Comment11: - same for Figure 7. Comparison of PSO-BBA (top) and BBA (bottom) generated adversarial examples (Query times and the distortion (L2 distance)). There are the results of your algorithm - explain them in text and emphasize the difference between the 24 images represented. Same for figure 6, explain the results more thoroughly in text.  Same comment for Figures 8 and 9. It would be also beneficial to compare the results obtained by your algorithm with similar ones from existing literature, validating thus the results.

Explanation: Thanks for the reviewer’s advice. The comments are all valuable and very helpful for revising and improving our paper. The experimental results are only compared with the same type of methods. The experimental results are described in the article. The difference between the 24 images is a subjective evaluation.

Comment12: -in the conclusion section please emphasize once again the advantages and disadvantages of your approach, also in comparison with similar work presented in the existing literature.  Al last but not at least please refine your English (grammar and style). The phrases are sometimes very short - sounding clipped. The grammar should be revised. There are even some funny expressions like " In the light of […], it come in two flavors:"(rows 54 and 60). Flavor refers to taste, not classification. A native English proof-reader might be of real help if you want to improve the intelligibility and quality of your work.

Explanation: Thanks for the reviewer’s advice. The comments are all valuable and very helpful for revising and improving our paper. We tried our best to improve the manuscript and made some changes in the manuscript. These changes will not influence the content and framework of the paper. And we marked them in red in revised paper. We appreciate for Editors/Reviewer’s hard work earnestly, and hope that the correction will meet with approval. Once again, thank you very much for your comments and suggestions.

Round 2

Reviewer 2 Report

This paper is worth for acceptance.

Author Response

Thanks for the reviewer’s advice. The comments are all valuable and very helpful. We appreciate for Reviewer’s hard work earnestly. Once again, thank you very much for your comments and suggestions.

Reviewer 3 Report

The paper quality has been significantly improved after revision.

You might want to increase the size of images in figures 6,7 and 8 in order to improve reader accessibility 

Author Response

Thanks for the reviewer’s advice. The comments are all valuable and very helpful for revising and improving our paper. We appreciate for Reviewer’s hard work earnestly, and hope that the correction will meet with approval. Once again, thank you very much for your comments and suggestions.